# Intermodal Four-Wave Mixing Process in Strain-Induced Birefringent Multimode Optical Fibers

**DOI:** 10.3390/ma15165604

**Published:** 2022-08-15

**Authors:** Michał Kwaśny, Paweł Mergo, Marek Napierała, Krzysztof Markiewicz, Urszula A. Laudyn

**Affiliations:** 1Faculty of Physics, Warsaw University of Technology, 75 Koszykowa, 00-662 Warsaw, Poland; 2Laboratory of Optical Fiber Technology, University of Maria Curie-Skłodowska, 3 M. Curie-Skłodowskiej Square, 20-031 Lublin, Poland; 3InPhoTech Sp. z o.o., 400 Poznańska, 05-850 Ołtarzew, Poland

**Keywords:** four-wave mixing, strain-induced birefringence, nonlinear frequency conversion

## Abstract

Our study investigated the partially degenerate intermodal four-wave mixing (IM-FWM) process in nonlinear multimode optical fibers with strain-induced birefringence. The difference in the refractive index along the two orthogonal directions was due to the photoelastic effect that occurred when the fiber under test (FUT) was subjected to uniformly applied diameter stress caused by winding on a cylinder of a given diameter. Our work analyzed how the nonlinear frequency conversion and the output modal field profiles depended on the degree of birefringence in FUT. The experimental results significantly affected the order of the excited moduli in fiber sections characterized by different amounts of birefringence. We also checked the efficiency of the FWM process for different polarizations of the pump beam to determine those for which the FWM process was most effective for the 532 nm sub-nanosecond pulses. More than 30% conversion efficiency was obtained for the FUTs with a length of tens of centimeters.

## 1. Introduction

The four-wave mixing (FWM) process of multimode optical fibers is a common technique that allows light parametric conversion through a third-order nonlinear process. The generated photon pairs that correspond to the Stokes and anti-Stokes waves primarily depend on the fiber’s dispersion properties. If the pump’s wavelength is close to the zero-dispersion wavelength, the phase-matching condition will determine the spectral separation of both waves generated in a nonlinear process. The first experimental reports date back to the 1970s [1], and since then, this method has been increasingly used for the nonlinear conversion of beam frequencies, particularly for single- [2,3,4] and multimode fiber systems [5,6,7,8,9], and also in photonic crystal fibers [10,11,12,13].

Conventional optical fibers are characterized by a relatively weak contrast of the refractive index between the core and cladding, and a bulk silica dispersion predominantly defines the zero-dispersion wavelength (ZDW), around 1.3 μm. As the dispersion can be modified over a certain range [14], the generation of shorter wavelengths using a single-mode approach is limited. One technique that attracted considerable interest realizes the FWM process in multimode fibers by utilizing an intermodal approach. It assumes the propagation of the pump and the Stokes and anti-Stokes modal fields in optical modes of different orders to fulfill the phase-matching condition [5,6,7,8,9,15,16]. Intermodal FWM (IM-FWM) has become a popular technique for converting the energy of the pump lasers to visible and infrared spectral regions. The IM-FWM approach extends the possibilities of the nonlinear systems and enhances the performance of parametric amplification and wavelength conversion systems. Additionally, in the emerging field of quantum communication science, the effective realization of FWM in optical fibers provides correlated photon pairs and contributes to the frequency conversion of quantum states of light [17].

In our previous study [9], we observed that due to fiber impurities, residual stress, and fabrication tolerances, the intermodal four-wave mixing process could be strongly dependent on coupling misalignments and polarization [18,19]. Moreover, the stability of the nonlinear frequency conversion and the output power of generated Stokes and anti-Stokes fields were susceptible to external factors. One of the benefits of such sensitivity is the utilization of nonlinear optical fibers in sensing applications [20,21]. Conversely, in laser applications [22,23], good stability, low sensitivity to external conditions, and high wavelength conversion efficiency in performing nonlinear process are crucial. An additional signal beam can be utilized to improve wavelength conversion efficiency within the IM-FWM process and realized in a few-mode nonlinear optical fiber [24]. In [9], we demonstrated a partially degenerated IM-FWM process with an effective generation of visible red and blue light from 532 nm sub-nanosecond pulses where an additional signal beam was generated within the same type of IM-FWM process. In some applications, utilizing a signal beam would be unfavorable due to the higher complexity of the optical system or to its higher cost. To provide a trade-off between the optical system complexity and operational efficiency, we analyzed the same type of optical fiber as in [9]; however, our choice for this study was characterized by a low optical birefringence induced by external strain. For this purpose, the fiber under test (FUT) was subjected to a uniformly applied diametral stress caused by winding on a cylinder of a given diameter. The proposed solution provided better isolation between the polarization degeneracy of the guided modes.

Consequently, we observed a stable and efficient IM-FWM process for the specific polarization of a fundamental beam. An additional advantage of our solution is the ability to easily control of the magnitude of the stress, which directly affects phase-matching for different nonlinear processes concerning the FWM phenomenon. Thus, controlling the orders of optical modes in which the Stokes/anti-Stokes beams are generated was possible by varying the winding diameter of the optical fiber.

## 2. Materials and Methods

### 2.1. Nonlinear Multimode Fiber

The nonlinear frequency conversion process involving the propagation of fundamental, Stokes, and anti-Stokes waves in the form of different order fiber modes was experimentally investigated in a graded-index optical fiber with a core diameter of 27 μm and a 0.008 refractive index contrast. The modal field diameter of the fundamental mode at 532 nm was about 7.6 μm. The effective index of refraction for the fundamental mode and a few higher-order ones were calculated and plotted in Figure 1a. A zero-dispersion wavelength (ZDW) corresponded to 1292 nm for the fundamental mode. With the exception of modes LP_11_, LP_02_, and LP_21_, other higher-order mode groups that correspond to visible wavelength bands vanish for wavelengths shorter than ZDW, as presented in Figure 1b. In a straight section of unstressed fiber, two effective nonlinear processes that result from the IM-FWM can be phase-matched. The Stokes/anti-Stokes beams can be generated and propagated in the form of LP_02_/LP_01_ modes at 616.8 nm/467.7 nm or LP_12_/LP_01_ modes at 642.2 nm/454.1 nm, respectively. Fiber bending induces birefringence proportional to the bending radius; thus, the phase-matching condition is also changed. The Stokes and anti-Stokes waves can propagate in the form of different modal groups or can be phase-matched at slightly different wavelengths.

### 2.2. Nonlinear Coupling Theory

The intermodal four-wave mixing in multimode optical fibers can be explained in terms of the nonlinear mode coupling [25,26]. In this case, the IM-FWM’s efficiency of the nonlinear coupling process depends on the phase-matching condition:(1)$$\Delta \beta_{kmpr}=\frac{n_{k}\left(\omega_{k}\right)\omega_{k}-n_{m}\left(\omega_{m}\right)\omega_{m}-n_{p}\left(\omega_{p}\right)\omega_{p}+n_{r}\left(\omega_{r}\right)\omega_{r}}{c}$$
and the mode-coupling coefficient:(2)$$Q_{kmpr}=\frac{{\left(\varepsilon_{0}cn_{eff}\right)}^{2}}{12}*\frac{\iint \left(F_{k}\cdot F_{m}^{*}\right)\left(F_{p}^{*}\cdot F_{r}\;\right)dxdy}{N_{k}N_{m}N_{p}N_{r}}$$
where *k*, *m*, *p*, and *r* enumerate fiber modes, *n_k_*(*ω_k_*) is the effective index of the *k*-th mode at optical frequency *ω**_k_*, *ε_0_* is the vacuum dielectric permittivity, *c* is the speed of light, *n_eff_*
*≈ n_k_*(*ω_k_*), and *F_k_*(*ω_k_*) is transverse mode distribution and a normalization term. In the presented notation, the modes {*k*, *r*} were chosen as pump modes; however, this choice is arbitrary due to process symmetry. We can assume that the mode field distribution does not depend on the wavelength; thus, the four-wave mixing process appears only for certain combinations of modes and their respective wavelengths involved in the process. The combinations of mode-order and its wavelength can be predicted by combining two conditions: the phase-matching Δ*β* = 0 and the energy conservation Δ*ω* = 0. Our next step involved calculating the mode-coupling coefficients for every phase-matched mode-wavelength combination, considering that only the combinations with the highest mode coupling coefficients will be involved in the nonlinear processes.

The initial four-wave mixing process starts with degenerate pumping at *ω_k_* = *ω_r_* = *ω_pump_* from LP_01_ mode. After that, different intermodal mixing processes appear simultaneously. Initial four-wave mixing leads to the cascading process, which involves fiber modes of higher orders and introduces additional wavelengths in the final spectrum [7]. Such processes simultaneously perturb the modal structure of the initial beam.

Fiber bending introduces birefringence and removes mode degeneracy. Thus, the polarity of the pumping mode becomes the additional optimization parameter. The FWM process does not guarantee polarization conservation during the cascading process. A complete map of possible processes, including polarization states, fiber modes, and phase-matching conditions, is tough to present concisely. Instead, in Figure 2, we show a simplified map of theoretically predicted mode-generation paths.

Since the utilized type of FUT supported 42 mode groups at 532 nm of pump wavelength, it was treated as multimode. For the experimental investigations, the FUT sections of 0.8 m were wound on spools differing in diameters to control the amount of stress-induced birefringence and determine the influence of fiber bending on the FWM process.

## 3. Results

The experimental setup is shown in Figure 3. As a source of intense optical pulses within a visible band (λ = 532 nm, 500 ps pulse width, 1 kHz repetition rate, and 45 kW peak power), we used the second harmonic of a Q-switched Nd:YAG laser, generated with the use of a KTP nonlinear crystal. A half-wave plate (λ/2) and a polarizer were used to control the optical power. An additional half-wave plate mounted in the motorized rotation stage (PRM1Z8) was in front of the coupling objective (10X Plan Achromat Objective, NA = 0.25) to precisely control the polarization of the pump beam. A pump beam was focused on a spot of a diameter w_0_ = 2.8 μm and coupled into the FUT. According to the adopted notation, the linear polarization direction parallel/perpendicular to the winding plane of the optical fiber was marked as 0°/90°, respectively. The experimental investigation of the IM-FWM processes in strain-induced birefringent multimode fibers involved the exact characterization of the optical output field for various polarization states of a pump beam. For that, we recorded the modal field distribution and the spectra of the output. To evaluate the performance of the nonlinear wave mixing, we monitored the power of the Stokes and anti-Stokes waves and then calculated the conversion efficiency as a percentage ratio of the power of Stokes and anti-Stokes waves to the power of the pump beam.

To evaluate the influence of birefringence on the IM-FWM process, we compared the experimental results with one obtained in a straight section of the FUT of the same type. We first checked the spectral characteristics of the pump beam to ensure that we were using a beam with a well-defined and narrow spectral distribution—Figure 4a. Next, to evaluate the IM-FWM process’s performance in the FUT, we recorded the spectral characteristics and modal field profiles for the anti-Stokes, pump, and Stokes waves for various optical powers of the pump. The experimental results are shown in Figure 4b. Assuming perfect coupling conditions, we predominantly excited the fundamental mode for a pump beam at 532 nm and observed the well-defined Stokes/anti-Stokes waves at 615.7 nm/469.0 nm, respectively. We saw that the four-wave mixing process was realized within the fundamental optical field and two LP_11_ modes. The effective energy transfer from the fundamental beam started from the average optical power of about 5.0 mW and saturated at about 10.0 mW (Figure 4c), leading to about a 20% frequency conversion efficiency. The analyzed IM-FWM process in a straightforward configuration allows for the utilization of generated beams as coherent and correlated light sources characterized by a shifted wavelength concerning the pump beam [27]. Our experimental setup is straightforward when expanded with an additional signal beam; as proposed in our previous study [9], the setup enables high conversion efficiency.

In the optical fibers’ production process, it is necessary to expect a particular parameter tolerance spread. In the case of nonlinear processes, even minor changes in the geometry or the distribution of the fiber’s refractive index can entail significant changes in the phase-matching conditions for the nonlinear process. Furthermore, each optical fiber exhibits some degree of heterogeneity that can cause many problems in obtaining the same efficiency in different samples of the same fiber type. Even a slight misalignment can reduce FWM efficiency or lead to phase-matching at different combinations of higher-order modes. In Figure 4d,e, we present an example of two nonlinear four-wave mixing processes leading to the generation of Stokes/anti-Stokes waves in which the pump beam was not coupled precisely enough and excited fundamental and some higher-order modes. In such a situation, we observed reduced wavelength conversion efficiency and/or more than one nonlinear process that led to the generation of more than one pair of FWM-based optical fields. Both situations are disadvantageous, for example, in the applications of fiber lasers or sensing devices.

Maintaining a straight and unstressed fiber of a length of about one meter in a commercial application would be impractical since an external pressure can cause a change in optical parameters, which directly impacts a nonlinear wave-mixing process. Such weakness can be exploited as an advantage since we can intentionally force a certain birefringence in the fiber, allowing better control over the IM-FWM process, increased stability, and better control of the order of generated Stokes/anti-Stokes optical fields. Our solution is to induce optical birefringence by mechanically straining the fiber, where the amount of birefringence can be controlled by the diameter of the cylinder on which the FUT is wound. Additional advantages of such a solution are the reduced size of the optical system and less vulnerability to the polarization degeneracy of the guided modes. It was therefore essential to investigate the IM-FWM process in strain-induced birefringent multimode optical fibers.

To induce birefringence in the fiber sections of 0.8 m, they were wound on spools of diameters equal to φ = 20 mm, φ = 25 mm, and φ = 30 mm. For this study, different amounts of birefringence were induced, and we determined the optimal conditions for the FWM process. The experimental results for the pump beam of a power *p* = 8.0 mW are presented in Figure 5. The photos in Figure 5a refer to the FUT of the winding diameter of φ = 30 mm, concerning the distribution of the anti-Stokes, pump, and Stokes beams (left, middle, and right panel, respectively), whereas the top, middle, and bottom rows refer to the polarization of the pump beam parallel (0°), tilted at 45°, and orthogonal to the winding plane (90°), respectively. The photos in Figure 5b,c obtained for the FUTs of φ = 25 mm and φ = 20 mm are organized identically. Different orders of the optical modes of the Stokes/anti-Stokes optical fields were generated, depending on the winding diameter. Minor differences were also seen for different pump beam polarizations. Figure 5d–f plots the spectral characteristics referring to the pump beam polarized at parallel (red line), tilted at 45° (orange line), and orthogonal to the winding plane of the FUT (red line). Contrary to the results obtained in a straight section of the FUT (Figure 4b), regardless of the winding diameter, one common feature was observed: in all cases, a pump beam excited a mix of the fundamental and higher-order modes, ensuring the possibility of adapting to the appropriate nonlinear IM-FWM process. Due to a low birefringence value of the wound FUT for some polarizations of the pump beam, we observed a distribution of the output field that was precisely assigned to a particular mode order. Among the results presented in Figure 5a–c, we can identify the system’s parameters (winding diameter/polarization of the pump), for which we obtained the most unambiguous distribution of the modal field of the Stokes/anti-Stokes beams generated as a result of one of the IM-FWM processes.

The most uniform optical field distributions at the output of the FUT were obtained for the winding diameter and pump polarization of φ = 30 mm/45° and φ = 20 mm/90°, respectively. In the cases mentioned above, the generated optical fields of anti-Stokes/Stokes waves propagated in the form of LP_11_/LP_31_ and LP_11_/LP_02_, respectively. Additionally, the spectral characteristics presented in Figure 5d–f indicate that two sharp peaks corresponded to the anti-Stokes/Stokes only for the winding diameters of φ = 30 mm and φ = 20 mm, respectively. Both IM-FWM peaks are considerably broader for the FUT wound on a cylinder of φ = 25 mm.

A more detailed characterization of the IM-FWM process that involves an arbitrary polarization of the pump beam was done with FUTs characterized by the winding diameter and pump polarization of φ = 30 mm/45° and φ = 20 mm/90°, respectively. Figure 6a–d show the spectral characteristics corresponding to the Stokes and anti-Stokes beam frequencies obtained for the pump polarized at 0°, 45°, and 90°, marked by blue, gray, and orange lines, respectively. For the φ = 30 mm cylinder, we noticed that the optimal polarization of the pump was 45° as the spectral width of both the Stokes and anti-Stokes peaks was the narrowest, compared with the blue and orange lines for 0° and 90° pump polarization, respectively. For the φ = 20 mm cylinder, we identified that the most spectrally narrow Stokes and anti-Stokes peaks were generated when the pump beam was polarized perpendicular to the winding plane (90°). Moreover, in this case, the power of the Stokes and anti-Stokes beams were the highest among all the measurement series, meaning they were also the most efficient frequency conversion within the IM-FWM process. The accurate characteristics showing the dependence of the output IM-FWM power as a function of the coupled power and the efficiency of the nonlinear process are shown in Figure 6e,f. Compared with the results obtained in a straight section of optical fiber (Figure 4c), we observed an increase in conversion efficiency of more than 30%. A higher efficiency was obtained due to the more selective power transfer from the pump beam to the Stokes and anti-Stokes beams. An additional advantage of using wound optical fiber is that it better maintains the order of the optical modal fields of the IM-FWM waves, which, in the case of the FUT of the φ = 20 mm winding diameter, remained in the LP_02_/LP_11_ for the Stokes and anti-Stokes beams within the pump polarization range of 45°–90°.

In order to verify the feasibility of the proposed frequency conversion method, we also checked the stability of the case presented in Figure 6e,f, characterized by the highest efficiency. For this purpose, we recorded the spectral characteristics over 20 min in two-minute increments, plotting the results in Figure 7a. We noticed that the wavelengths of both peaks at λ = 472.0 nm and λ = 610.8 nm corresponded to the anti-Stokes/Stokes wavelengths overlapping during the analyzed period, which proved their frequency stability. The issue of the IM-FWM waves’ power stability was more complex. Figure 7b shows that the power of the Stokes/anti-Stokes beams oscillates significantly over time, around the mean value of P_S_ = 0.774 mW (σ_S_ = 0.044 mW) and P_a-S_ = 1.502 mW (σ_a-S_ = 0.121 mW), where σ denotes a standard deviation of data presented in the plot in Figure 7b. The source of such significant changes in the power of the generated beams was predominantly the very low stability of the pump beam, which was obtained as the second harmonic beam in an additional simple optical system without using any elements to monitor and stabilize its power. In such a system, the low stability of the infrared laser that pumped KTP crystal in a nonlinear manner became apparent in the power fluctuation of the second harmonic. Indeed, for the λ = 532 nm beam, calculating the mean value of the power and its standard deviation, we obtained P_532nm_ = 8.022 mW (σ_532nm_ = 0.070 mW). This result meant that the difference between maximal and minimal pump power within the analyzed period was on the order of 0.3 mW, which directly translated to the variation in the power of the Stokes/anti-Stokes waves. In order to clearly show the dependence of the IM-FWM beams’ power versus time dependence, Figure 7c presents an enlarged plot fragment from Figure 7b. We see that a variation of the power of Stokes and anti-Stokes waves corresponded to an analogous change in the power of the pump beam, indicating a moderately strong relationship between the measured optical signals. Therefore, we concluded that using an invariant pump beam would significantly improve the power stability of the Stokes and anti-Stokes beams.

## 4. Conclusions

Optical fibers have a relatively low fabrication cost and great freedom in the design of optical parameters. Few-mode and multimode optical fibers can be used effectively as nonlinear mediums in the optical frequency conversion process. By managing strain-induced birefringence, nonlinear fibers can be used as a medium for full-fiber light sources characterized by controllable modal field distribution. In the dynamic change of fiber stress, the distribution of the Stokes and anti-Stokes fields could also be used to determine the amount of pressure on the fiber. However, our results of the utilization of wound optical fiber provides compact solutions for stable operating devices designed for wavelength conversion systems and in the field of quantum optics rather than optical sensing.

## Figures and Tables

**Figure 1 materials-15-05604-f001:**
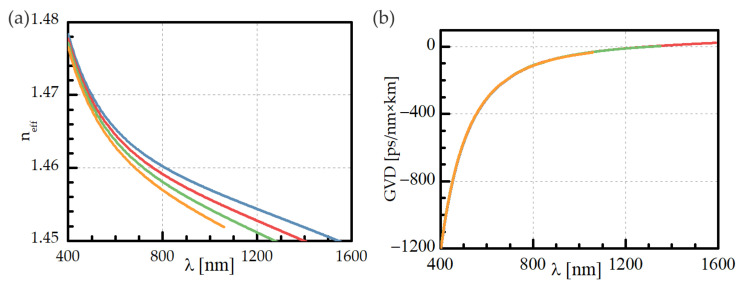
(**a**) The effective refractive index value, plotted for fundamental mode (blue) and a few higher-order modes: LP_11_ (red), LP_02_ and LP_21_ (green), and LP_31_ and LP_12_ (orange); (**b**) group velocity dispersion (GVD) for different modes: LP_01_ and LP_11_ (light red), LP_02_ and LP_21_ (green), and LP_12_ and LP_31_ (orange).

**Figure 2 materials-15-05604-f002:**
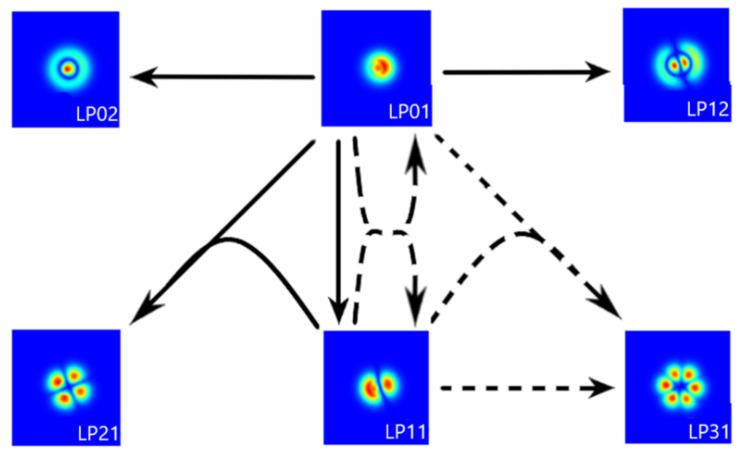
A simplified map of the theoretical four-wave mixing mode-generation path represents the theoretical LP_xx_ modes distributions in fiber with a 30 mm bending radius. A single arrow represents a degenerated four-wave mixing process, where both pumping beams have the same modal structure. Solid/dashed lines represent transitions with conserved/mixed polarization states, respectively. The second generated beam (not shown) has the same modal structure as one of the pump beams.

**Figure 3 materials-15-05604-f003:**
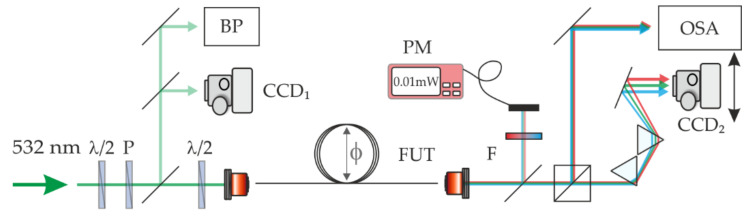
The experimental setup for the IM-FWM investigation in a strain-induced birefringent multimode fibers: λ/2—half-wave plate, P—polarizer, FUT—fiber under test, BP—beam profiler, PM—power meter, F—spectral filter, CCD—digital camera, and OSA—optical spectrum analyzer.

**Figure 4 materials-15-05604-f004:**
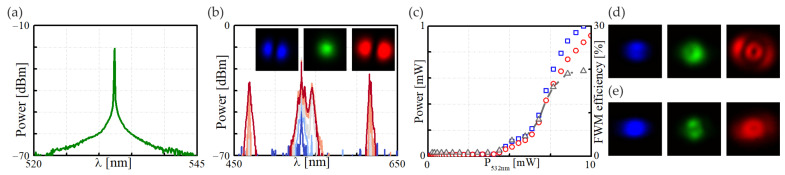
The four-wave mixing process in a 0.8 m, stress-free FUT. Spectral characteristics: (**a**) the pump beam (*p* = 0.1 mW and λ = 532 nm) and (**b**) the optical output field (plotted for the pump beam powers of 0.5 mW, 4.0 mW, 5.0 mW, 6.0 mWb and 9.0 mW–dark blue to solid red lines). The insets present a modal field distribution of (from the left) an anti-Stokes, pump, and Stokes beam: (**c**) the power of Stokes (red circles) and anti-Stokes (blue squares) matched with the IM-FWM efficiency (gray triangles) as a function of the power of a pump beam; (**d**,**e**) the modal field distribution at the FUT’s output in the case of sub-optimal beam coupling: (from the left) the anti-Stokes, pump, and Stokes beams.

**Figure 5 materials-15-05604-f005:**
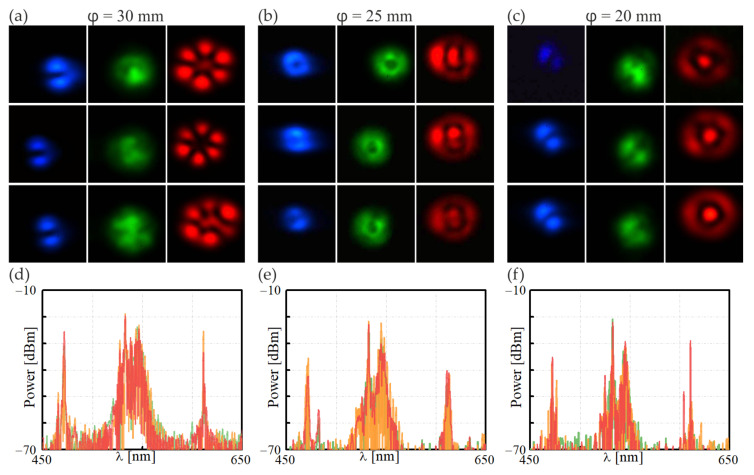
Anti-Stokes, pump, and Stokes optical field distribution (**left**, **middle,** and **right** panels, respectively) obtained in the IM-FWM process (P_532nm_ = 8.0 mW) in FUTs with different amounts of optical birefringence and as a result of winding fiber on spools of varying diameters: (**a**) φ = 30 mm, (**b**) φ = 25 mm, and (**c**) φ = 20 mm. Each sub-section’s top, middle, and bottom panels refer to the polarization of a pump beam: parallel, tilted at 45°, and orthogonal to the winding plane, respectively; (**d**–**f**) spectral characteristics correspond to the panels (**a**–**c**) for the pump beam polarized at 0° (green line), 45° (orange line), and 90° (red line), respectively.

**Figure 6 materials-15-05604-f006:**
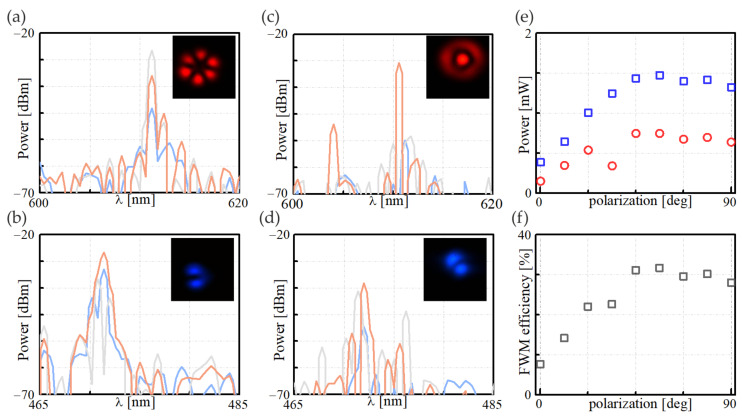
A close-up view of the spectral characteristics of the Stokes and anti-Stokes beams generated in the IM-FWM process in FUT of a winding diameter: (**a**,**b**) φ = 30 mm and (**c**,**d**) φ = 20 mm, for the pump polarization of 0°, 45°, and 90° (blue, gray, and orange lines, respectively). The insets present the Stokes/anti-Stokes modal profiles for the pump polarization: (**a**,**b**) 45° and (**c**,**d**) 90°; (**e**) the Stokes (red circles) and anti-Stokes (blue squares) beams’ power and (**f**) efficiency of the IM-FWM process that corresponds to (**c**,**d**), plotted for the pump beam power of 8.0 mW.

**Figure 7 materials-15-05604-f007:**
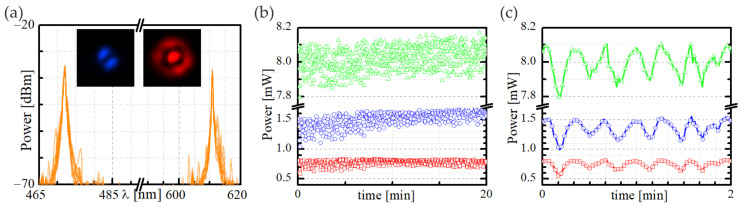
Stability of the IM-FWM process in wound FUT (φ = 20 mm) for the pump beam polarized orthogonal to the winding plane: (**a**) spectral characteristics recorded over 20 min in two-minute increments. The insets present the Stokes (**left**) and anti-Stokes (**right**) modal field profiles; (**b**) the power of the Stokes (red squares), anti-Stokes (blue circles), and pump beam (green triangles) within 20 min; (**c**) an enlarged section of the graph (**b**).

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
