# Peer review of "Intermodal Four-Wave Mixing Process in Strain-Induced Birefringent Multimode Optical Fibers"

_materials, 2022, doi:10.3390/ma15165604_

Round 1
Reviewer 1 Report
In this manuscript the authors present an intermodal four-wave mixing process in stress-induced multimode fiber experimentally. The conversion efficiency is improved, and this phenomenon and operation principle are explained with the recorded mode distribution. As Figure 5 presents, the birefringence strongly depends on the diameters of fiber spools, and different birefringence value gives different intermodal FWM. Therefore, I think the stability of the output power and spectral of signal and idler lights is in a poor state, while it is important and essential for practical applications. I don’t recommend this manuscript could be published on Materials in its current state unless much more evidences with enough novelty could be present.
Author Response
Dear reviewer,
Thank you very much for the detailed review of our manuscript. We addressed suggested comments in the reviewed version of the draft. To clarify the purpose of this work, I would like to indicate that our previous work [1] focused on the inter-modal FWM process in graded-index optical fiber and its optimization with an additional signal beam. In strain-induced birefringent multimode fibers we present a strong dependence on the phase-matching to a particular nonlinear wave mixing process on the value optical birefringence. Thus, using a much simpler measurement setup, we optimize IM-FWM efficiency to provide a trade-off between optical system complexity and operational efficiency, obtaining more than 30% conversion efficiency.
Addressing the concerns about the stability of the generated beams, we carefully verified the stability of the IM-FWM process in wounded FUT and confirmed that spectral stability is at a remarkably high level. We also explained that the poorer temporal stability is mainly due to the instability of the pump beam power. Additional Figure (number 7) and paragraph was added at the end of the Results section.
We hope the posted explanations are sufficient to explain all the doubts and ambiguities noted while reading our work.
Regards,
Michal Kwasny
[1] Kwaśny, M.; Mergo, P.; Napierała, M.; Markiewicz, K.; Laudyn, U.A. An Efficient Method for the Intermodal Four-Wave Mixing Process. Materials 2022, 15, 4550. https://doi.org/10.3390/ma15134550
Reviewer 2 Report
In this manuscript, the authors have presented a system for the intermodal four-wave mixing process in multimode fibers. To improve the performance of the system, a cylinder is used to apply diametral stress to the fiber. The concept has been validated by experimental results with providing a relatively high conversion efficiency. However, there are some points regards the proposed design which would be better to be discussed and investigated:
11. It should be clearly mentioned that the work is based on a previous recent work of the same authors [1]. A clear paragraph in the introduction should be devoted to the novelty of the manuscript compared to this previous work.
[1] Kwaśny, M.; Mergo, P.; Napierała, M.; Markiewicz, K.; Laudyn, U.A. An Efficient Method for the Intermodal Four-Wave Mixing Process. Materials 2022, 15, 4550. https://doi.org/10.3390/ma15134550
22. Related to the previous point, winding the fiber under test seems to be one of the main differences compared to the previous work, however, does it really helpful, the currently achieved efficiency is 30% in the best case while the previously obtained one was 40%? The used lengths for both systems are indeed different but fair comparisons (for fibers with the same length) can provide a clear clue about the enhancement between the two systems.
33. The manuscript does not present a satisfactory level of background in the introduction. More state-of-the-art solutions should be briefly mentioned with concluding the main research gaps that the authors are trying to address by such a proposed system.
44. To clarify the contribution of the proposed system, more quantitative and qualitative comparisons should be added. For example in terms of conversion efficiency, the ease of achieving phase matching, implementation complexity, robustness against alignment tolerances, and so on.
55. Achieving the phase matching condition should be a critical requirement, however, there is no discussion in the paper about how it can be guaranteed especially in the existence of misalignment and fabrication tolerances.
66. It would be interesting for the readers if the authors can collaborate more about how and why this specific approach can be used for sensing applications.
Author Response
Dear reviewer,
Thank you very much for the detailed review of our manuscript. We addressed suggested comments in the reviewed version of the draft. We would also like to provide clarification on the questions given:
- It should be clearly mentioned that the work is based on a previous recent work of the same authors [1]. A clear paragraph in the introduction should be devoted to the novelty of the manuscript compared to this previous work.
[1] Kwaśny, M.; Mergo, P.; Napierała, M.; Markiewicz, K.; Laudyn, U.A. An Efficient Method for the Intermodal Four-Wave Mixing Process. Materials 2022, 15, 4550. https://doi.org/10.3390/ma15134550
The part of the description comparing the obtained results with those described in [1] has been expanded and commented on. We indicated that the presented configuration could be a trade-off between optical system complexity and operational efficiency.
- Related to the previous point, winding the fiber under test seems to be one of the main differences compared to the previous work, however, does it really helpful, the currently achieved efficiency is 30% in the best case while the previously obtained one was 40%? The used lengths for both systems are indeed different but fair comparisons (for fibers with the same length) can provide a clear clue about the enhancement between the two systems.
Our intention with this article was to present that realizing the IM-FWM process does not always demand an extensive apparatus system. Compared to our previous article [1], we present a stable and efficient inter-modal wave mixing process that occurs for the specific polarization of a fundamental beam. At the expense of a slight decrease in efficiency, we obtain a very compact system that supports easy control of the orders of generated optical modes within the IM-FWM process. Thus, controlling the orders of optical modes in which the Stokes/anti-Stokes beams are generated is possible by varying the winding diameter of the optical fiber.
- The manuscript does not present a satisfactory level of background in the introduction. More state-of-the-art solutions should be briefly mentioned with concluding the main research gaps that the authors are trying to address by such a proposed system.
We expanded the introduction section and explained the proposed solution for the efficient and stable experimental realization of the four-wave mixing using an inter-modal approach.
- To clarify the contribution of the proposed system, more quantitative and qualitative comparisons should be added. For example in terms of conversion efficiency, the ease of achieving phase matching, implementation complexity, robustness against alignment tolerances, and so on.
To expand the analytical part of the work, we verified the stability of the IM-FWM process in wounded FUT and confirmed that spectral stability is at a remarkably high level. We also explained that the poorer temporal stability is mainly due to the instability of the pump beam power. Additional Figure (number 7) and paragraph was added at the end of the Results section.
- Achieving the phase matching condition should be a critical requirement, however, there is no discussion in the paper about how it can be guaranteed especially in the existence of misalignment and fabrication tolerances.
It is difficult to include fabrication tolerances in proposed systems; however, when the system is once aligned, it exhibits a high-level of spectral and power stability. Changing optical parameters to a small extent can be compensated by using a cylinder of a different diameter.
- It would be interesting for the readers if the authors can collaborate more about how and why this specific approach can be used for sensing applications.
The presented system seems to be more promising in wavelength conversion systems or the field of quantum optics rather than optical sensing. Of course, the wounded fibre is susceptible to external pressure, but there are more precise methods for strain sensing.
Reviewer 3 Report
The paper deserves to be published. There are just a few minor changes and questions to be addressed. Please see the attached pdf file.

Author Response
Dear reviewer,
Thank you very much for the detailed review of our manuscript. All the comments in the reviewed manuscript (lines 67, 162, 227) were addressed and commented/corrected in the revised version. Regarding the question about the text fragment in verse 209, we did not check precisely the smaller diameters of the spool. However, in our first tests, we also checked what the process efficiency looks like for a bend radius of about 12mm, and it was much lower compared to the results presented in this article.
We hope the posted explanations are sufficient to explain all the doubts and ambiguities noted while reading our work.
Regards,
Michal Kwasny
Round 2
Reviewer 2 Report
The authors successfully addressed all the comments. I believe that the paper level is enhanced significantly after the revision, especially the introduction part, so I recommend the acceptance in its current form.
Author Response
Dear Reviewer,
Thank you again for the detailed review of our manuscript and appreciating the corrections in the draft. In addition, minor language errors have been corrected in the uploaded latest version of the manuscript.
Regards,
Michal Kwasny